# Study on the Mechanism of Influencing Adolescents’ Willingness to Participate in Ice Sports

**DOI:** 10.3390/children10061080

**Published:** 2023-06-19

**Authors:** Zhuoling Liu, Sai Wang, Qian Gu

**Affiliations:** 1School of Physical Education, Shandong University, Jinan 250100, China; liuzl678@163.com; 2School of Economics and Management, Hebei University of Science and Technology, Shijiazhuang 051432, China; wangsai98516@163.com

**Keywords:** S-O-R theory, MOA-TAM integrated model, influence mechanism, adolescent

## Abstract

Background: Ice sports are an effective means to promote the physical health of adolescents, and the willingness to participate in ice sports is the basis of adolescents’ awareness of their participation in ice sports and an important prerequisite for their participation. The aim of this study is to investigate the mechanisms influencing adolescents’ willingness to participate in ice sports. Methods: Using the stimulus-organism-response theoretical framework, the motivation-opportunity-ability model, and the technology acceptance model, a theoretical model of the influencing mechanisms of adolescents’ participation in ice sports was constructed. A total of 3419 secondary school students were surveyed, and the data were analyzed using structural equation modeling, the mechanisms influencing adolescents’ willingness to participate in ice sports and the moderating role of perceived riskiness and perceived ease of use are proposed and tested. Results: The study found that participation motivation, participation ability, perceived usefulness, and perceived risk all have significant effects on adolescents’ willingness to participate in ice sports. The degree of influence in descending order is as follows: ability to participate, perceived usefulness, motivation to participate, and perceived risk. Perceived risk plays a mediating role in the process of motivation to participate and ability to participate influencing willingness to participate. Perceived ease of use plays a mediating role in the process of motivation to participate influencing perceived usefulness. Conclusions: This study provides a systematic understanding of the mechanisms that influence adolescents’ willingness to participate in ice sports. The findings provide valuable insights into the subject of adolescents’ ice sport participation and can inform the development of strategies to increase participation in these activities. Future research should deepen the investigation of the patterns of willingness to participate in ice sports behavior among adolescents, which is important for promoting the sustainable development of ice sports, improving the health of adolescents, and advancing the construction of a healthy China, a healthy world, and global health.

## 1. Introduction

The results of the China Student Physical Health Monitor in recent years show that there are numerous physical health problems among adolescents and the detection rate of obesity continues to rise [1,2,3,4]. Ice sports, also known as snow sports, are a variety of sports played on ice and snow, for example: skiing, curling and other sports. Participation in ice sports can effectively improve the physical health problems of adolescents. It is a good way for adolescents to maintain their weight, improve their courage to face difficulties and overcome them, and is very beneficial to their physical quality [5,6,7]. Under the influence of the 2022 Beijing Winter Olympic Games, the development of ice sports in China has gained momentum, driving adolescents to actively participate in ice sports, promoting the formation of adolescents’ ice hobby circle and providing a reserve of talent for subsequent Winter Olympic Games’ preparations. So how do we further increase adolescents’ participation in ice sports? The answer is obvious: only if adolescents have a real desire to participate in ice sports can they develop the habit of exercising in ice sports [8]. The willingness of adolescents to participate in ice sports is the basis of their awareness and an important prerequisite for their participation in ice sports [9,10].

Many scholars at home and abroad have conducted studies on adolescents’ participation in ice sports, taking the family, school and social levels as the entry point for research, and have studied the value and influencing factors of adolescent participation in ice sports from the perspective of physical exercise [11,12,13,14,15,16,17,18,19], but fewer scholars have studied adolescents’ participation in ice sports from the behaviorist level of cognitive psychology [20,21]. The willingness of adolescents to participate in ice sports is a subjective assessment of the likelihood of their future participation in ice sports, and the predictive effect of willingness to participate on sporting behavior has been validated by many studies [22]. In this study, adolescents’ intention to participate in ice sports refers to the behavioral intention of adolescents to participate directly or indirectly in ice sports, such as participating in ice sports such as skiing and skating or participating in ice tourism or ice safety education seminars. Since the success of the 2022 Winter Olympics, ice sports have become a global phenomenon. Studies have confirmed that ice sports are an effective means of promoting physical health in adolescents and have a positive effect on preventing psychological problems, reducing the incidence of obesity, and preventing disease [23,24,25]. Willingness to participate in ice sports is an important prerequisite for determining whether adolescents participate in ice sports.

This study integrates sports science with behavioral science, psychology, sociology and other disciplines, introduces the SOR theoretical framework [26], and incorporates the MOA model and the TAM model into the stimulus (S) and organism (O) parts of the framework, respectively [27,28,29,30]. Based on behavioral pattern theory (MOA) and sociological theory (TAM), this study explores the influencing mechanism of Chinese adolescents’ willingness to participate in ice sports, further determines the path relationship among the factors that affect the willingness of adolescents to participate in ice sports, determines the current fundamental problems and causes, and puts forward corresponding suggestions. In this way, while improving the theoretical research on adolescents’ participation in ice sports, it also provides a certain reference value for the current government and stakeholders on how to create and develop better forms of adolescents’ participation in ice sports and launch more attractive ice sports products.

### 1.1. Theoretical Foundation

The S-O-R theoretical model was proposed by Mehrabian and Russell in 1974 [26]. The model suggests that an individual’s willingness to participate in sport is stimulated by internal and external drivers [31], leading to psychological, emotional, and cognitive changes that result in a state outcome of either pro or avoidance [32]. Thus, the S-O-R model can be used as a dynamic expression of an individual’s willingness to participate in sport [33]. In recent years, the S-O-R model has been applied to research in the field of sport participation Willingness. Adolescents’ participation in ice sports is also stimulated by internal and external drivers, leading to changes in their ice sports participation behavior and eventually resulting in the output of sport participation Willingness. Therefore, the S-O-R theoretical framework is an appropriate choice as the theoretical basis for studying the influence of adolescents’ participation in ice sports.

The MOA model is an analytical framework for explaining the occurrence of individual behavioral intentions, distilled and generalized by Macinnis and Jaworski on the basis of previous research [28]. The MOA model was initially used to analyze individual responses to information and the processing of behavior. As an integrated and flexible model, the model has been used by disciplines in numerous fields to study individual behavioral intentions, mainly in public and social management, social capital, human resource management and knowledge management, and in the field of sport mainly to study individual sport participation behavior [30,34,35,36,37,38], verifying the stability of the model and predictability of behavior.

The Technology Acceptance Model (TAM) is a model proposed by Davis in 1989. The model considers individual willingness as an external variable that influences individual behavioral willingness through two structural factors: perceived ease of use and perceived usefulness. While many scholars have conducted theoretical and empirical research around the Technology Acceptance Model (TAM), the results show that the model is only 40% accurate in predicting individuals’ willingness to engage in acceptance behaviors [29], and different variables need to be introduced in different domains to refine the model [39,40,41]. In the field of sports, the model is widely used to study willingness to exercise, willingness to consume sports, willingness to use sports platforms, etc. The ice sports in this study have a certain technical threshold, and individuals’ perceived risk can cause their willingness to participate in them [42,43]. Therefore, this study looks at the external environment variables (MOA) and the three latent variables of perceived ease of use, perceived usefulness, and perceived riskiness acting on adolescents’ willingness to participate in ice sports.

To sum up, this paper adopts the S-O-R theoretical model to investigate the psychological state of adolescents’ participation in ice sports under external drivers. However, only external environmental factors are considered in the “S” framework, which is insufficient [44]. Therefore, the MOA model, which considers motivational, opportunity, and ability factors, is introduced to provide a more comprehensive analysis of the internal and external drivers of individual behavior. The “O” framework of the MOA model is utilized as a mediating variable to capture the changes in individuals’ psychological states due to the internal and external drivers. To enhance the accuracy of the research, the TAM model is refined by introducing perceived riskiness, given the inherent risks in ice sports. Three psychological states, namely perceived usefulness, perceived ease of use, and perceived riskiness, are selected to represent individuals’ emotions and cognition in the TAM model.

### 1.2. Research Hypothesis

To enhance the accuracy and comprehensiveness of the TAM model, this study integrates the MOA model as the external variable for the TAM model. The TAM model alone can only explain 40% to 60% of individual behavior due to its unstable external variables [45,46]. In contrast, the MOA model provides a more systematic and in-depth analysis of both internal and external drivers of individual behavior. Specifically, the MOA model’s motivation, opportunity, and ability factors directly impact an individual’s willingness to act, and as such, it serves as the “S” part of the S-O-R framework in this study. It is worth noting that the relationship between “S”, “O”, and “R” is not a simple linear progression from “S” to “O” and then to “R”. Instead, research has shown that there is a more complex “S” → “O” → “R” relationship, particularly at the stimulus level where “S” has a direct impact on “R”.

Stimulus (S)—Motivation (M)

The motivation for adolescents’ participation in ice sports is rooted in the individual motivation of adolescents’ to be physically active. As their individual motivation increases, so does their willingness to participate in sports. Shu conducted an empirical study based on the TAM model, using motivation as an external variable. The study concluded that participation motivation has a significant positive effect on willingness to participate, and that participation motivation also positively influences perceived usefulness and perceived ease of use [47]. Additionally, Zhang and other scholars found through empirical research that the risk of sports injury is a factor that influences adolescents’ ice sports participation behavior. Interestingly, the study found that the stronger an individual’s motivation to participate in a certain behavior, the lower their assessment of perceived riskiness [48,49]. Based on the above theoretical analysis, this study proposes the following hypotheses.

**H1.** 
*Adolescents’ motivation to participate in ice sports has a positive influence on their intention to participate in ice sports.*


**H2.** 
*Adolescents’ motivation to participate in ice sports has a positive influence on their perceived usefulness of ice sports.*


**H3.** 
*Adolescents’ motivation to participate in ice sports has a positive influence on their perceived ease of use of ice sports.*


**H4.** 
*Adolescents’ motivation to participate in ice sports has a negative influence on their perceived riskiness of ice sports.*


The opportunity to participate in ice sports is an essential factor that affects adolescents’ willingness to engage in these activities. Li et al. found that the availability of venues and adequate facilities positively influence the willingness of adolescents to participate in ice sports [50,51]. In the current environment, however, adolescents’ participation in ice sports remains low due to a lack of concrete support measures and social forces. As a result, the systematic mobilization and long-term development of adolescents’ ice sports must be considered [52,53].

This study proposes the following hypothesis based on the above analysis:

**H5.** 
*Opportunities for adolescents’ participation in ice sports positively influences their willingness to participate in ice sports.*


**H6.** 
*Opportunities for adolescents’ participation in ice sports have a positive effect on their perceived usefulness of ice sports.*


**H7.** 
*Opportunities for adolescents’ participation in ice sports have a positive impact on their perceived ease of use in ice sports.*


**H8.** 
*The availability of opportunities for adolescents’ participation in ice sports negatively influences their perceived riskiness of ice sports.*


Stimulus (S)—Ability (A)

In addition, adolescents’ ability to participate in ice sports is a crucial factor in their willingness to participate. When adolescents possess the appropriate knowledge base and professional skills to cope with injuries [54], they perceive ice sports as more useful and easier to engage in, reducing their perception of risk [55]. Based on this, the following hypotheses are proposed:

**H9.** 
*Adolescents’ ability to participate in ice sports positively influences their perceived usefulness of ice sports.*


**H10.** 
*Adolescents’ ability to participate in ice sports positively influences their perceived ease of use in ice sports.*


**H11.** 
*Adolescents’ ability to participate in ice sports negatively influences their perceived riskiness of ice sports.*


**H12.** 
*Adolescents’ ability to participate in ice sports positively influences their willingness to participate in ice sports.*


Perceived ease of use is a significant factor in the Technology Acceptance Model (TAM) and has been found to have a positive effect on perceived usefulness in previous studies [56,57]. Additionally, when adolescents perceive ice sports as enjoyable and easy to participate in, this can positively influence their willingness to participate. Perceived riskiness, on the other hand, is defined as the degree of uncertainty regarding the outcome of adolescents’ participation in ice sports and is an important factor to consider given the technical difficulty and potential danger of ice sports.

Based on these considerations, the following research hypotheses are proposed:

**H13.** 
*Adolescents’ perceived ease of use of ice sports positively influences their perceived usefulness of ice sports.*


**H14.** 
*The perceived usefulness of ice sports positively influences adolescents’ willingness to participate in ice sports.*


**H15.** 
*Perceived ease of use of ice sports among adolescents’ positively influences their willingness to participate in ice sports.*


**H16.** 
*The perceived riskiness of ice sports negatively influences adolescents’ willingness to participate in ice sports.*


In summary, this study proposes a theoretical model of the mechanisms influencing adolescents’ participation in ice sports, drawing on previous research and theoretical findings (Figure 1).

## 2. Materials and Methods

### 2.1. Design and Participants

In order to test the feasibility of the questionnaire and to revise and define the formulation of the questions, the study began with a small pre-survey, with 206 questionnaires returned. The pre-survey questionnaire was then analyzed for reliability and validity to test its reliability and validity. After the pre-survey questionnaire passed the reliability and validity tests, the language expressions of individual questions in the questionnaire were improved through the respondents’ feedback and suggestions. The full questionnaire is attached at Appendix A. To ensure the representativeness of the sample, a combination of stratified and random sampling was used to select a total of 3552 secondary school students from three provinces and cities in China, namely Beijing, Tianjin, and Hebei. The three provinces of Beijing, Tianjin, and Hebei were chosen for the study because of their geographical proximity and the abundance of snow and ice resources. The study protocol was approved by the ethical committee at Wuhan Sports University (ethical code: 2023022). Each participant was asked to indicate their willingness to participate in this study by signing an informed consent form; those who refused to participate in the study were excluded. Data were collected and analyzed anonymously. The study was conducted in accordance with the ethical requirement of the Declaration of Helsinki.

The Influencing Mechanisms of Adolescents’ Participation in Ice Sports Scale was administered to adolescents’ using a combination of online and offline methods. A total of eleven secondary schools in three locations were selected for offline data collection, and adolescents were invited to fill in the questionnaire at the entrance of schools, with gifts provided as incentives. Online data collection was carried out using Questionnaire Star and study groups of educational institutions. The survey was conducted from December 2022 to February 2023. After excluding questionnaires that took less than one minute to complete and those that were incomplete, 3419 valid questionnaires were collected, resulting in an effective rate of 96.3%.

### 2.2. Measurement

#### 2.2.1. Physical Activity Motivation Measure (MPAM-R) 

The questionnaire was developed by Frederick and Ryan [58], the questionnaire used in this study is a 15-item questionnaire translated and simplified by Chinese scholar Chen. The questionnaire measures adolescent’s motivation to participate in ice sports from a multidimensional perspective, classifying motivation into health motivation, fun motivation, ability motivation, appearance motivation and social motivation. The reliability of the questionnaire is 0.958, which meets the measurement requirements. 

#### 2.2.2. Exercise Condition Measure 

Using the exercise conditions survey in Qiu et al.’s study of university students’ exercise behavior, a unidimensional scale consisting of three questions, with higher scores indicating that the adolescent perceives his or her sporting conditions and environment to be more acceptable was developed [59]. The scale has good construct validity and a reliability of 0.771.

#### 2.2.3. Scientific Exercise Knowledge Scale 

The design of the questions on an adolescent’s participation in ice sports is based on the research of Zhao and contains two dimensions of ice sports knowledge and ice sports skills, with a total of four questions [60]. This part of the scale has a reliability of 0.913 and good construct validity.

#### 2.2.4. Perceptual Measurement Questionnaire 

Adopted from Davis’s perceptual measurement item, the 9-question scale consists of three dimensions, which measure the perceived usefulness, perceived risk, and perceived ease of use of an adolescent’s participation in ice sports [55]. The scale has good reliability and validity and meets the requirements of measurement.

#### 2.2.5. Willingness to Participate in the Movement Scale Questionnaire 

The measure of adolescent’s willingness to participate in ice sports was conducted using the Adolescent Sport Behavioral Intentions Scale from Jiang and Li’s study [61]. The reliability of the scale was 0.727, which indicates that the scale used has good internal consistency.

The questionnaire items were scored on a 5-point Likert scale, with 1–5 being “strongly disagree—strongly agree”. The scale has good reliability and validity and meets the requirements of measurement. The Cronbach alpha coefficients for the subscales ranged from 0.727 to 0.958, with an overall alpha coefficient of 0.956, and the scales used had good internal consistency.

### 2.3. Statistics and Processing of Data

The questionnaire data were entered into SPSS 22.0 and validity analysis, reliability analysis and exploratory factor analysis were performed on the entered data. In addition, the confirmatory factor analysis was performed using AMOS 26.0 on the entered data. The overall Cronbach’s alpha coefficient for this study’s questionnaire was 0.956, indicating high reliability. As presented in Table 1, the Cronbach’s alpha coefficient values for each subscale were all above 0.7, indicating high reliability of each subscale. The average variance extracted (AVE) values were all above 0.5, indicating good convergent validity of the questions within the latent variable. The composite reliability (CR) values were also above 0.7, indicating high internal consistency of the test questions for each subscale. The KMO value of the questionnaire data in this study was 0.974, and the significance of the Bartlett sphericity test was less than 0.0001. The significant value of the Bartlett’s sphericity test was 0.0001, indicating that the variables were correlated with each other and suitable for factor analysis.

In this study, the correlation coefficients and AVE roots of each variable were analyzed and the arithmetic square root of the AVE corresponding to each variable was greater than 0.5, and the absolute value of each variable’s correlation coefficient with the other variables was less than the square root of the AVE of that variable, indicating good discriminant validity among the variables in this study (Table 2).

## 3. Results

### 3.1. Preliminary Analysis

The sample consisted of 1322 male students and 2097 female students, with 1936 junior high school students and 1583 senior high school students. The detailed data are shown in Table 3. Descriptive statistical analysis of the scale by importing the questionnaire data into SPSS 22.0 showed that the grades under different genders follow an approximate normal distribution.

### 3.2. Hypothesis Testing

The structural model in this study was tested using AMOS 26.0 software and the test results showed a good fit (GFI = 0.869, RMSEA = 0.090, RMR = 0.035, CFI = 0.873, NNFI = 0.860) and the corresponding structural equation model was constructed as Figure 2.

In this study, observed variables were used to measure motivation to participate, opportunity to participate, ability to participate, perceived ease of use, and willingness to participate. As a result, some indicators are correlated, leading to a covariate relationship between some of the residuals. To account for this, the initial structural equation model was modified based on the modified index MI and the related path relationships. Specifically, covariance relationships were added between e1 and e8, e1 and e7, e2 and e6, e13 and e14, e16 and e33, e11 and e33, and e13 and e14. The overall fitness of the model was improved after the modification, with the following fit indices: GFI = 0.897, RMSEA = 0.032, RMR = 0.015, CFI = 0.901, and NNFI = 0.890. Ultimately, the revised model was accepted for this study. The revised structural equation model is shown in Figure 3, and the corresponding path coefficients and hypothesis testing results are presented in Table 4.

In the pathway relationship diagram for adolescents’ motivation to participate in ice sports, one pathway fails the test, three pathways pass the test, Hypothesis 2 does not hold, and Hypotheses 1, 3 and 4 hold. The table of revised model path coefficients shows that three of the path relationships hypotheses for adolescents’ ice sports participation opportunities passed the test and one did not, so Hypotheses 6, 7 and 8 are valid and Hypothesis 5 is not valid. All four pathway relationship hypotheses for adolescents’ participation in ice sports passed the test, with Hypotheses 9, 10, 11 and 12 all holding true. One of the two pathway relationships for the perceived ease of use of adolescents’ participation in ice sports failed the test, Hypothesis 13 was valid and Hypothesis 14 was not. The pathway relationship hypotheses for perceived usefulness and perceived riskiness of adolescents’ ice sports participation both passed the test, so Hypotheses 15 and 16 hold.

### 3.3. Agency Analysis

The structural equation modeling of the mechanisms influencing adolescents’ participation in ice sports constructed in this study passed the AMOS test. The Bootstrap method was used to investigate potential mediating effects among motivation to participate, opportunity to participate, ability to participate, perceived ease of use, perceived usefulness, perceived risk, and willingness to participate. The results are presented in Table 5. The table indicates that the total, direct, and indirect effects of perceived risk are all significant within the 95% confidence interval, with direct effects accounting for 81.9% and 91.2%, respectively, and indirect effects accounting for 18.1% and 8.8%, respectively. 

These findings suggest that perceived risk plays a partial mediating role in the process by which motivation and ability to participate affect willingness to participate. Meanwhile, the total, direct, and indirect effects of perceived ease of use were significant within the 95% confidence interval, with direct effects accounting for 96.1% and indirect effects accounting for 3.9%. This suggests that perceived ease of use plays a partial mediating role in the process by which participation motivation influences perceived usefulness. However, the indirect effect of perceived ease of use was not significant at the 95% confidence interval, and the direct effect accounted for 99.7% of the effect, indicating that the direct effect plays a dominant role in the process by which opportunity to participate influences perceived usefulness. 

## 4. Discussion

The results of the study show that the willingness of adolescents to participate in ice sports is influenced by multiple factors. In this study, the influence degrees of key variables on adolescents’ willingness to participate in ice sports from high to low are as follows: participation ability, perceived usefulness, participation motivation, and perceived risk. Perceived risk plays an intermediary role in the process of participation motivation and participation ability influencing participation intention, and perceived ease of use plays an intermediary role in the process of participation motivation influencing perceived usefulness. The findings of this study provided support for the validity of a majority of the hypotheses, as 13 out of the 16 hypotheses were supported. 

### 4.1. The Influence of Participation Ability on Participation Willingness

The ability to participate in ice sports is one of the most important factors influencing the willingness of adolescents to participate in ice sports. Some scholars have shown in their studies that the ability to participate is one of the most important factors affecting the willingness to participate [62]. Furthermore, the findings of this study provide evidence that adolescents’ willingness to participate in ice sports is positively influenced by their ability to participate, both directly and indirectly through the mediating effect of perceived risk. The mean values of the 14 indicators of participation ability reveal that adolescents have limited skills and knowledge of ice sports, indicating the need to improve their ability to participate in such activities.

Specifically, the study highlights the significant role of participation ability, perceived risk, motivation to participate, and perceived usefulness in influencing adolescents’ willingness to engage in ice sports. Ability is the knowledge, skills, and experience that an individual has to perform a particular behavior and complete a particular task. It reflects the knowledge and skills associated with a behavior [63]. The results suggest that enhancing participation ability can have a direct positive impact on participation willingness, as well as an indirect positive impact through the mediating effect of perceived risk. The reason for this may be that when individuals have some knowledge of ice sports and sports injury protection, this will increase their awareness of ice sports and reduce their perception of risk [64,65], thus promoting their willingness to participate in ice sports.

Overall, the findings underscore the importance of developing and implementing interventions aimed at improving adolescents’ participation ability in ice sports, as well as reducing perceived risk, and enhancing motivation and perceived usefulness to promote greater participation willingness.

### 4.2. The Effect of Perceived Usefulness on Willingness to Participate

The Technology Acceptance Model (TAM) identifies users’ perceived usefulness and perceived ease of use of new things as key factors influencing participants’ behavioral intentions [29]. The results show that perceived ease of use had a significant positive impact on perceived usefulness, so the greater the perceived ease of use, the higher the perceived usefulness, and the stronger the intention to participate. This result is also in line with Davis et al.’s findings that users’ willingness to participate is influenced by usefulness and ease of use, with perceived ease of use indirectly influencing willingness to participate through the mediation of perceived usefulness [66]. The reasons for this are twofold: on the one hand, adolescents can feel the charm and value of ice sports in the process of participating in ice sports, and their attitude towards ice sports will be more positive when they perceive that ice sports exercise can keep them healthy, release academic pressure and enrich their spare time; on the other hand, among the mechanisms that influence adolescents’ willingness to participate in ice sports, perceived ease of use will indirectly influence participation. Therefore, the more convenient the process is, the easier it is to use the equipment or the quicker they can master the technical movements of a sport, the more positive they feel about ice sports and the more willing they are to participate.

### 4.3. The Influence of Participation Motivation on Participation Willingness

Studies have pointed out that there is a significant positive correlation between participation motivation and perceived ease of use [67,68]. In addition, some researchers have found that participation motivation has a positive impact on participation intention, which is basically consistent with our research results [26]. The results of the modified structural equation model demonstrated that the stronger the motivation of adolescents to participate in ice sports, the higher their perceived ease of use, the lower their perceived risk, and the stronger their intention to participate. This finding is consistent with the research of Deci and Ryan, as well as Vallerand and Losier [69,70]. Social psychologists generally support exercise motivation to help explain an individual’s involvement in an activity or behavioral persistence, or as a relationship between stability and persistence. The findings of this study provide evidence that adolescents’ willingness to participate in ice sports is significantly influenced by motivation and perceived usefulness. Notably, participation motivation was found to indirectly influence the motivation to participate through perceived risk, highlighting the key role of participation motivation in driving adolescents’ willingness to engage in ice sports, which is consistent with prior research conducted by Weinberg and Gould [71]. Additionally, the results indicate that perceived usefulness is influenced by both participation ability and motivation, suggesting that adolescents’ willingness to engage in ice sports is strengthened when they possess the necessary skills and knowledge to participate and are motivated to do so.

### 4.4. The Effect of Perceived Risk on Participation Willingness

The structural equation modelling and mediating effect results show that the perceived risk has a significant negative effect on adolescents’ participation in ice sports. Yu et al. show that if users’ perceived risk of participation is reduced, the more willing they are to participate [72]. This is consistent with the findings of our study. In addition, it also plays a partial mediating role in the process of participation motivation and participation ability affecting participation in ice sports. Differences in individual judgement of potential risk can increase the likelihood of an individual developing a sports injury, and perceived risk is an important factor influencing an individual’s willingness to exercise. The present study provides evidence that the perceived riskiness of ice sports significantly and negatively impacts adolescents’ willingness to participate, and partially mediates the relationship between motivation to participate, ability to participate, and willingness to participate. Our findings are consistent with Bandura’s study, which suggests that perceived risk is a crucial factor and a strong predictor of sport participation [73].

### 4.5. Strength and Limitations

The strength of this study lies in the construction of a structural equation model that more systematically examines the influencing mechanisms of adolescents’ willingness to participate in ice sports, explores the possible mediating effects in the model, and further explains the mediating role of perceived riskiness and perceived ease of use in influencing adolescents’ willingness to participate in ice sports. However, the factors influencing adolescents’ willingness to participate in ice sports are complex, and the factors interact with each other and are constantly changing. Therefore, there are some limitations in this study. First, the theory is not systematic, and this research is still in the preliminary stage. The second limitation is about research results. In this study, adolescents were used as the research object to investigate the influencing mechanism of adolescents’ willingness to participate in ice sports through a questionnaire survey, and to analyze the influencing factors of adolescents’ participation in ice sports hospitals. However, caution must be exercised when interpreting these results, and some limitations should be kept in mind. 

Future research on adolescents’ willingness to participate in ice sports could continue to be improved in several ways: firstly, by expanding the sample size. More than just an expansion of the sample size, follow-up studies could increase sample diversity by incorporating feedback from participants in other sports similar to ice sports into the sample. Secondly, subsequent studies could review scales from previous studies on motivation, opportunity, ability and other related variables and then select appropriate scales according to the topic to be studied and could improve the scales by adding perceived recreation through expert evaluation and in-depth interviews, in order to better fit the topic of the study.

## 5. Conclusions

This study reveals that adolescents’ willingness to participate in ice sports is influenced by their ability to participate, motivation to participate, perceived usefulness and perceived risk, and is a complex system of multiple influences. The most significant factor found in the structural equation model was adolescents’ ability to participate in ice sports. Considering these findings, it is recommended that efforts be made to increase adolescents’ motivation to participate in ice sports, address their safety concerns about the activity, implement systematic programs to enhance adolescents’ ability to participate, and guide them in developing positive habits and behaviors towards ice sports. By taking these steps, we can promote the sustainable development of adolescents’ ice sports.

## Figures and Tables

**Figure 1 children-10-01080-f001:**
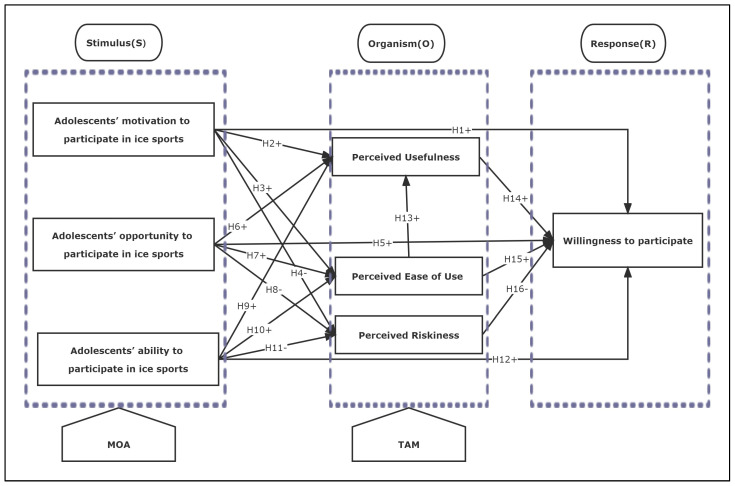
Theoretical research model of the influence mechanism of adolescents’ participation in ice sports.

**Figure 2 children-10-01080-f002:**
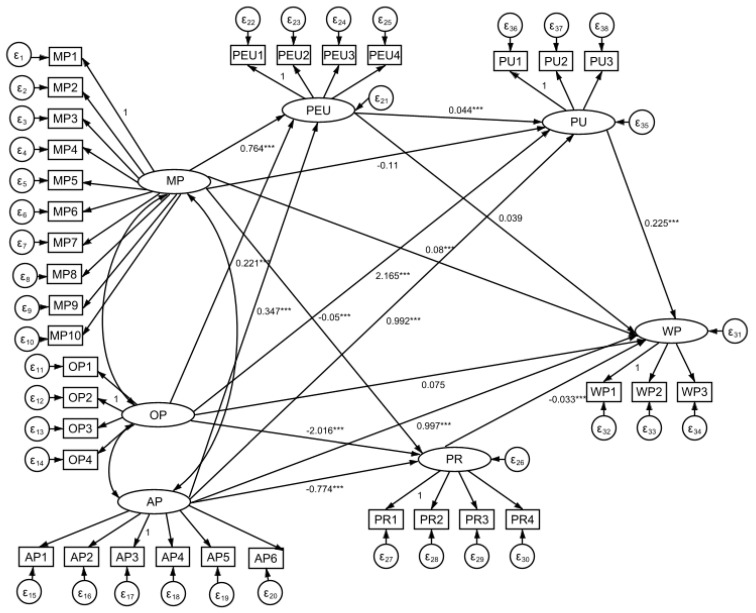
Structural equation model construction diagram of the influence mechanism of adolescents’ participation in ice sports. Note: MP = Motivation to Participate; OP = Opportunity to Participate; AP = Ability to Participate; PU = Perceived Usefulness; PEU = Perceived Ease of Use; PR = Perceived Riskiness; WP = Willingness to Participate. *** referring to *p*-value under 0.01.

**Figure 3 children-10-01080-f003:**
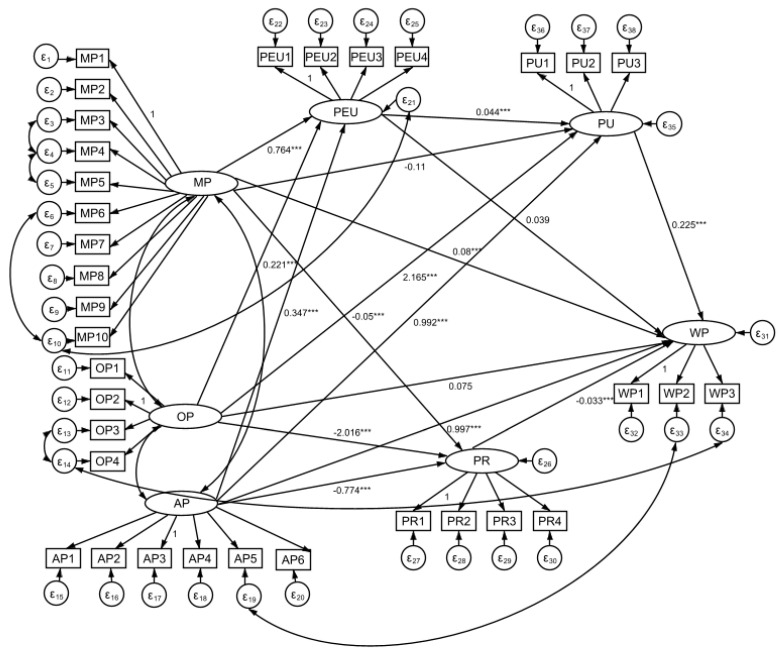
Revised structural equation model construction diagram. Note: MP = Motivation to Participate; OP = Opportunity to Participate; AP = Ability to Participate; PU = Perceived Usefulness; PEU = Perceived Ease of Use; PR = Perceived Riskiness; WP = Willingness to Participate. *** referring to *p*-value under 0.01.

**Table 1 children-10-01080-t001:** Results of reliability and validity analysis of the questionnaire.

Variables		Cronbach’s α	AVE	CR	KMO	Bartlett’s Spherical Test
Motivation to Participate	Health Motivation	0.958	0.712	0.832	0.962	<0.0001
	Ability Motivation		0.846	0.916		
	Interest Motivation		0.774	0.872		
	Appearance Motivation		0.665	0.799		
	Social Motivation		0.618	0.763		
Opportunity to Participate		0.771	0.683	0.895	0.841	<0.0001
Ability to Participate	Knowledge Base	0.913	0.622	0.832	0.915	<0.0001
	Physical Skills		0.665	0.856		
Perceived Ease of Use		0.865	0.586	0.703	0.87	<0.0001
Perceived Usefulness		0.957	0.85	0.958	0.868	<0.0001
Perceived Riskiness		0.850	0.62	0.862	0.746	<0.0001
Willingness to participate		0.727	0.571	0.701	0.732	<0.0001

**Table 2 children-10-01080-t002:** Pearson correlation coefficient and AVE root value.

	1. Health Motivation	2. Ability Motivation	3. Interest Motivation	4. Appearance Motivation	5. Social Motivation	6. Opportunity to Participate	7. Knowledge Base	8. Physical Skills	9. Perceived Ease of Use	10. Perceived Usefulness	11. Perceived Riskiness	12. Willingness to Participate
1	0.844											
2	0.792 **	0.92										
3	0.803 **	0.867 **	0.88									
4	0.792 **	0.844 **	0.848 **	0.815								
5	0.757 **	0.754 **	0.828 **	0.796 **	0.786							
6	0.321	0.292	0.317	0.327	0.345	0.826						
7	0.296	0.256	0.288	0.308	0.326	0.704 **	0.789					
8	0.286	0.229	0.269	0.283	0.306	0.735 **	0.84 **	0.815				
9	0.348	0.321	0.334	0.362	0.355	0.74 **	0.697 **	0.688 **	0.609			
10	0.717 **	0.69 **	0.766 **	0.746 **	0.862 **	0.363	0.359	0.349	0.377	0.922		
11	0.342	0.316	0.334	0.344	0.365	0.897 **	0.731 **	0.735 **	0.532 *	0.369	0.787	
12	0.655 *	0.634 *	0.704 **	0.694 **	0.735 **	0.578 *	0.636 *	0.629 *	0.595 *	0.723 **	0.605 *	0.735

Note: ** and * represent the 5% and 10% significance levels respectively, and the diagonal numbers are the root values of the AVE for that factor.

**Table 3 children-10-01080-t003:** Effective sample personal basic information table.

Statistical Variables	Attributes	Number of People (N)	Percentage (%)
Gender	Male	1322	38.67%
	Female	2097	61.33%
Grade	First year of junior school	611	17.87%
	Second year of junior school	235	6.87%
	Third year of junior school	1090	31.88%
	First year of high school	1114	32.58%
	Second year of high school	369	10.79%
Family financial situation	Very well-off	10	0.29%
	More well-off	27	0.79%
	Average	2256	65.98%
	Difficult	945	27.64%
	Very difficult	181	5.30%

**Table 4 children-10-01080-t004:** Revised model path coefficient table.

Path Relationship			Path Coefficient	Standard Error	*p*-Value	Result
Motivation to Participate	→	Perceived Usefulness	−0.114	0.212	0.071	Failure to pass
Motivation to Participate	→	Perceived Ease of Use	0.764	0.016	***	Pass
Motivation to Participate	→	Perceived Riskiness	−0.05	0.01	***	Pass
Motivation to Participate	→	Willingness to participate	0.08	0.012	***	Pass
Opportunity to Participate	→	Perceived Usefulness	2.165	0.101	***	Pass
Opportunity to Participate	→	Perceived Ease of Use	0.221	0.101	**	Pass
Opportunity to Participate	→	Perceived Riskiness	−2.016	0.086	***	Pass
Opportunity to Participate	→	Willingness to participate	0.075	0.208	0.109	Failure to pass
Ability to Participate	→	Perceived Usefulness	0.992	0.084	***	Pass
Ability to Participate	→	Perceived Ease of Use	0.347	0.065	***	Pass
Ability to Participate	→	Perceived Riskiness	−0.774	0.07	***	Pass
Ability to Participate	→	Willingness to participate	0.997	0.105	***	Pass
Perceived Ease of Use	→	Perceived Usefulness	0.044	0.008	***	Pass
Perceived Ease of Use	→	Willingness to participate	0.039	0.128	0.217	Failure to pass
Perceived Usefulness	→	Willingness to participate	0.225	0.012	***	Pass
Perceived Riskiness	→	Willingness to participate	−0.033	0.073	***	Pass

Note: ** referring to *p*-value under 0.05, *** referring to *p*-value under 0.01.

**Table 5 children-10-01080-t005:** Results of a mediating effect test on the mechanisms influencing the willingness of adolescents to participate in ice sports.

Type of Effect	Path Relationship	Efficacy Value	LLCI	ULCI	*p*-Value	Percentage (%)
Total effect 1	Motivation to Participate → Willingness to Participate	0.830	0.232	0.934	0.000	
Direct effect 1	Motivation to Participate → Willingness to Participate	0.680	0.063	0.840	0.000	0.819
Indirect effect 1	Motivation to Participate → Perceived Riskiness → Willingness to Participate	0.150	0.085	0.215	0.000	0.181
Total effect 2	Ability to Participate → Willingness to Participate	0.921	0.893	1.179	0.000	
Direct effect 2	Ability to Participate → Willingness to Participate	0.840	0.909	1.184	0.000	0.912
Indirect effect 2	Ability to Participate → Perceived Riskiness → Willingness to Participate	0.081	0.004	0.178	0.000	0.088
Total effect 3	Ability to Participate → Perceived Usefulness	0.916	0.806	1.071	0.000	
Direct effect 3	Ability to Participate → Perceived Usefulness	0.880	0.778	1.010	0.000	0.961
Indirect effect 3	Ability to Participate → Perceived Ease of Use → Perceived Usefulness	0.036	0.024	0.054	0.000	0.039
Total effect 4	Opportunity to Participate → Perceived Usefulness	0.903	0.942	1.318	0.000	
Direct effect 4	Opportunity to Participate → Perceived Usefulness	0.901	0.813	1.086	0.000	0.997
Indirect effect 4	Opportunity to Participate → Perceived Ease of Use → Perceived Usefulness	0.002	−0.024	0.009	0.258	0.003

## Data Availability

The data presented in this study are available on request from the corresponding author.

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
