# Peer review of "Study on the Mechanism of Influencing Adolescents’ Willingness to Participate in Ice Sports"

_children, 2023, doi:10.3390/children10061080_

Round 1
Reviewer 1 Report
The present paper is pertinent and is well-designed. Moreover, it is seemed to be relevant to highlight the premise of this manuscript is a worthy one, and the authors spent a great time in the developing and structuring. However, there are few details/suggestions that need to be addressed to the manuscript prior to publication.
Introduction
The authors are invited to add references that support the idea exposed in the first paragraph.
The last paragraph of the introduction is very confused. The authors exposed the aim, then informed about the number of participants in the survey, and exposed other purpose. This last paragraph should be rewritten.
Results
To highlight the pertinent results, it is suggested to add a * referring to p-values under 0.05 or ** when p-values were under 0.001 (example in table 5).
Discussion
The authors suggested that future research should continue to expand the scope of the investigation. However, the sentence is modest and is unable to translate the need to solve future problems in this area. Can the authors highlight more needs for future research?
Reviewer 2 Report
DEAR AUTHORS:
After read this first review, some changes are proposed:
Line 32: Intention to participate in ice sports.
INCLUDE ONLY KEY WORDS
Line 35-44: INCLUDE SCIENTIFIC REFERENCES
Ice SPORTS: INCLUDE ABREVIATURE TROUGHT THE TEXT
Line 68-70: SOR, MOA, TAM: Include references in all these proposals
Line 212: I DON'T UNDERSTAD WELL PRE-SURVEY. Define better...
Line 230: Ethical Issues. Helsinki declaration?, University approval?
Line 264: INCLUDE CORRELATIONS AND MAGNITUDE OF CORRELATIONS (HOPKING)
Line 284: approximated a normal distribution, thus indicating the representativeness of the sample. DID YOU CHECK THE NORMALITY OF DATA BEFORE. AND WHICH METHOD DID YOU USE. EXPLAIN IT.
Line 291: validity between the variables in this study (Table 3). BETTER AMONG THESE
Line 355: Table 5: percentage, include %
Discussion part: IN THE FIRST PARAGRAPH INCLUDE MAIN RESULTS.
The study revealed that the higher the perceived ease of use of adolescents’ ice sports participation, the higher their perceived usefulness and willingness to participate. Conversely, the higher the perceived risk of adolescents’ ice sports participation, the lower their willingness to participate. AGREE WITH THIS INTERPRETATION, BUT CONTRAST WITH THE LITERATURE AND EXPLAIN THE POTENTIAL REASONS
The results suggest that enhancing participation ability can have a direct positive impact on participation willingness, as well as an indirect positive impact through the mediating effect of perceived risk. YES, BUT WHY, GIVE EXPLANATION…
INCLUDE STRENGHTS AND PRATICAL APPLICATIONS
FINALLY, THIS IS ONLY ONE SUGGESTION. I WOULD INCLUDE MORE 20-23 REFERENCES
MINOR CHANGES
Round 2
Reviewer 2 Report
DEAR AUTHORS:
AFTER CHECK THIS FIRST REVIEW. THE ARTICLE IS ACCEPTED
MODERATE